# Gene Amplification in Tumor Cells: Developed De Novo or Adopted from Stem Cells

**DOI:** 10.3390/cells12010148

**Published:** 2022-12-30

**Authors:** Ulrike Fischer, Eckart Meese

**Affiliations:** Institute of Human Genetics, Saarland University, Building 60, 66421 Homburg, Germany

**Keywords:** gene amplification, re-replication, differentiation, CDK4, CDT1

## Abstract

Gene amplifications have been known for several decades as physiological processes in amphibian and flies, e.g., during eggshell development in *Drosophila* and as part of pathological processes in humans, specifically in tumors and drug-resistant cells. The long-held belief that a physiological gene amplification does not occur in humans was, however, fundamental questioned by findings that showed gene amplification in human stem cells. We hypothesis that the physiological and the pathological, i.e., tumor associated processes of gene amplification share at their beginning the same underlying mechanism. Re-replication was reported both in the context of tumor related genome instability and during restricted time windows in *Drosophila* development causing the known developmental gene amplification in *Drosophila*. There is also growing evidence that gene amplification and re-replication were present in human stem cells. It appears likely that stem cells utilize a re-replication mechanism that has been developed early in evolution as a powerful tool to increase gene copy numbers very efficiently. Here, we show that, several decades ago, there was already evidence of gene amplification in non-tumor mammalian cells, but that was not recognized at the time and interpreted accordingly. We give an overview on gene amplifications during normal mammalian development, the possible mechanism that enable gene amplification and hypothesize how tumors adopted this capability for gene amplification.

## 1. Early Evidence of Gene Amplifications in Mammalian Cells during Differentiation

As early as 1968, studies reported features of gene amplification in normal mammalian cells. These first indications were, however, not followed up upon. Gene amplification, i.e., an increase in the gene copy number as a tool for an increased RNA and protein synthesis in differentiating tissues was first postulated in *Nature* by Pelc SR [1].

The first detailed experimental evidence for gene amplification stems from a study of Banerjee and Wagner, who analyzed mouse mammary gland from virgin, pregnant, lactating and weaned mice [2]. Prior to mammary gland isolation, DNA was radioactive labeled in vivo with ^3^H-dT and purified by CsCl gradient centrifugation. Surprisingly, the buoyant density pattern was different between the DNA from 6-day lactating mammary mice and the DNA from virgin, pregnant or weaned mice. The authors observed an additional labeled band near the heavier region of the gradient for lactating cells and interpreted these results as indication for a high number of actively replicating cells. Since the extra DNA was not found in the involuted mammary glands, the authors hypothesized a periodic of amplification occurrence in parts of the genome during differentiation of mammary gland [2]. Notably, earlier studies of the authors already mentioned this idea. These studies showed an abrupt rise in DNA synthesis at day 6 during differentiation without a corresponding increase in mitosis followed by a rapid decline of DNA synthesis [3]. In this context, an abrupt rise in DNA synthesis probably mirrors gene amplifications at this time point and the rapid decline afterwards probably a controlled stop of the gene amplification process.

In 1978, Hernández and colleagues determined reassociation kinetics of fragmented DNA from sperm and leucocytes at various time points [4]. They reported that sperm DNA contained a lower number of unique sequences than the corresponding leucocyte DNA (59% vs. 64%). The authors found a sharp change in the curve of C_0_t values in spermatozoa and interpreted this observation not only as indication of highly repetitive DNA but also as indication of “periodic” gene amplification in actively differentiating cells [4].

Additionally, in 1978 Strom and Dorfman reported evidence for amplification during differentiation in chicken cartilage and postulated that these sequence amplifications were regulatory and “turn on” the synthesis of differentiation-specific genes without replication of the whole genome [5]. In detail, they found a 3.6-fold reassociation difference between differentiating chicken neural retina and a corresponding not differentiated retina sample and 2.0-fold reassociation difference between differentiating chicken cartilage and a not differentiated cartilage sample. The authors speculate that the difference in reassociation between both experiments stems either from differences of amplification frequency between both tissue types and from differences of amplification frequency of several cell types within a tissue-like neural retina. This is where it may have been recognized for the first time that the detection of gene amplification in a cell population depends on the purity of a probe [5].

Another study provides evidence for an actin gene amplification during chicken myogenesis. In 1981, Schwartz and Rosenblum speculated that an increased mRNA expression of the chicken actin gene during myogenesis is controlled at the transcriptional level [6]. An addendum to their manuscript described unpublished Southern hybridization experiments demonstrating a 10–100-fold increase in the alpha-actin gene that may be responsible for the increase in alpha-actin mRNA. The amplified actin DNA disappeared at later myogenic stages [6]. The latter results were only published as meeting report in a book in 1982 [7].

Ten years later, a study reported the enrichment of double minutes after addition of serum to serum-free mouse embryo (SFME) cells. However, the meaning of double minutes as manifestation of gene amplification and a characteristic marker was not recognized in this context [8,9]. The abnormal karyotype was related to immortalization or interpreted as an escape from crisis of the SFME-cells but not as gene amplification [8]. At that time, it was not clear that mouse serum-free embryo cells were mostly neural stem-like cells, and that the addition of serum leads to differentiation accompanied by gene amplification on double minutes.

## 2. Evidence of Gene Amplification during Differentiation of Stem Cells

Once it has been recognized that serum-free mouse embryo (SFME) cells represent largely neural stem cells, gene amplification were searched for in mouse and human neural stem cells using more sensitive methods. Array comparative genomic hybridization (arrayCGH) detected amplified regions from 250 kb to more than 10 Mb in size with a genome-wide distribution in human and mouse neural stem cells during differentiation including in the above mentioned SFME cells [10,11]. Further analysis using fluorescence-in situ-hybridization revealed amplifications on a single cell level in 5–33% of the investigated cell population. Fluorescence in situ hybridization (FISH) also revealed a heterogeneous pattern of amplification with co-amplification of genes varying between cells [11]. The moderate and varying overall amplification level found by FISH was also reflected by a moderate increase in log_2_ values between 0.16 and 0.35 in the arrayCGH experiments.

Further and more detailed evidence of gene amplification was found by studies that employed three different induction protocols to differentiate neural stem cells into either astrocytes, oligodendrocytes, or neurons. The pattern of gene amplification during these differentiation processes were exemplarily analyzed for eight genes that were previously found to be amplified in arrayCGH analysis of differentiating neural stem cells. *CDK4* was amplified during the differentiation into astrocytes, oligodendrocytes, and neurons. *MDM2* was amplified in most of the differentiation protocols but not during the first three days of differentiation towards oligodendrocytes. *GINS2* was amplified only after one day of differentiation towards oligodendrocytes and neurons but not towards astrocytes [12]. These data indicate that amplification is highly specific for different differentiation processes.

Further example of gene amplification during differentiation and placentation stems from a study by Meinhardt and colleagues who analyzed extravillous trophoblast (EVT) differentiation. They detected *ERBB2* amplifications by dual color silver in situ hybridization (SISH) in human first trimester placental tissue sections [13]. Specifically, they found 3–6 signals for *ERBB2* in 29% of HLA-G^+^ CC trophoblasts [13]. Using a centromere chromosome 17 probe the authors excluded whole chromosome gains.

Selective amplifications were also detected in parietal trophoblast giant cells of mouse placenta. Whole genome sequencing (WGS) and digital droplet PCR identified five amplified regions including some that mapped on mouse chromosome 13. Interestingly each of the amplified regions on chromosome 13 contain different placenta gene families [14]. Although this study is not on stem cell differentiation it shows gene amplification connected to developmental processes comparable to the developmental gene amplifications that was long known for *Drosophila*.

Another example for gene amplification stems from studies on human mesenchymal stem cells (hMSCs) and from mesenchymal stem cells differentiating towards adipocytes and osteoblasts [15]. ArrayCGH identified 12 amplified chromosome regions in undifferentiated hMSCs, 18 amplified chromosome regions during adipogenic differentiation and 19 amplified chromosome regions during osteogenic differentiation of hMSCs [15]. Most amplified chromosome regions were different from those regions that have previously been described as amplified in neural stem cell differentiation or trophoblast differentiation. While these findings argue for a high specificity of these amplifications, there were also gene amplifications such as the amplification of *CDK4* and *MDM2* that were more generally found not only in hMSCs, but also during adipocyte and osteoblast differentiation, and during neural stem cell differentiation. The analysis of gene amplification in human mesenchymal stem cells also shed light on the challenge of amplification detection. While whole genome sequencing (WGS) detected many amplified chromosome regions, it does not allow to identify all amplifications that were detected by arrayCGH. For example, gene amplification of *CABIN1* gene in undifferentiated hMSCs was detected by arrayCGH and on single cell level by FISH but not by WGS [16]. These examples show that amplification detection largely depends on the used techniques and that the derived data should be carefully interpreted.

In addition, during myoblast differentiation gene amplifications were detected. In detail, myoblast differentiation from mononucleate cells towards multinucleate myotubes was investigated on primary human myoblasts and the mouse C2C12 myoblast cell line by qPCR experiments with TagMan probes [17]. Specifically, *CDK4*, *MDM2* and *NUP133* were investigated for amplification during the primary human myoblast differentiation and *ACTA1*, *CDK4*, *MYO18B* and *NUP133* were investigated during the C2C12 myoblast cell differentiation. While, human primary myoblasts showed amplification of *CDK4*, *MDM2* and NUP133, C2C12 myoblast cells showed *CDK4* amplification already in undifferentiated cells and the highest level of amplification during differentiation. This analysis also exemplified the challenge of identifying specific gene amplifications. Alpha-actin (*ACTA1*) gene amplification was not detected during C2C12 differentiation using qPCR with two different *ACTA1* TagMan probes [17]. This is surprising since alpha-actin gene amplification was supposed to account for elevated expression levels during chicken myogenesis [6]. QPCR analysis, however, detected amplification of the *NUP133* gene, which localizes only 10 kb next to *ACTA1*. Since a genomic probe (BAC-clone) that encompassed *ACTA1* and *NUP133* for FISH analysis, revealed amplification, it is very likely that the hybridization probe used in former studies on chicken myogenesis included sequences from *NUP133* gene and that the hybridization results were misleadingly interpreted as *ACTA1* gene amplification. In Figure 1 main methods for detecting gene amplification were explained.

## 3. Concordance of Gene Amplifications in Stem Cells, Stem Cell Differentiation and Tumors

As detailed above, gene amplifications have been known in tumors for several decades and were more recently also reported for both undifferentiated and differentiating stem cells. We summarize similarities of gene amplifications in stem cells and their corresponding malignant tumors.

ArrayCGH did identify amplification on 93 chromosome regions in normal human neural progenitor (NHNP) cells differentiated for 5 days [10]. The gained chromosome regions include regions on chromosome 12 with the genes *CDK4* and *MDM2*. QPCR analysis revealed *EGFR* amplification in NHNP cells and in differentiated NHNP cells [10,12]. Glioblastoma that are the most malignant primary brain tumors with astrocytic origin [18], showed amplification of the *EGFR* gene in about one third of the cases [19]. In about 15% of glioblastoma cases the *CDK4* gene is amplified and in around 10% the *MDM2* gene [20,21]. In addition to the above-mentioned genes, 19 amplified chromosome regions identified by arrayCGH in neural progenitor cells were likewise identified as gained chromosome regions in a study on TCGA (The Cancer Genome Atlas) glioblastoma samples [22].

As reported above during extravillous trophoblast (EVT) differentiation *ERBB2* amplification was detected [13]. Likewise, *ERBB2* amplification was reported in gestational trophoblastic disease (GTN) that is a rare group of tumors that were mostly benign and origin from placental villous and extravillous trophoblast cells [23,24,25] *ERBB2* amplification and expression in combination with hyperploidy leads to a higher proliferation and more aggressive behavior including metastases and carcinoma [23].

FISH analysis revealed co-amplification of CDK4 and MDM2 during mesenchymal stem cell differentiation towards adipocytes [15]. Well-differentiated liposarcoma (WDL) and dedifferentiated liposarcoma (DDL) were both associated with high-level amplifications of CDK4 and MDM2 [26]. Among adipocytic tumors that were the most common mesenchymal neoplasms, WDL lack metastatic potential but 10% can dedifferentiate to DDL with aggressive local growth and increased tendency for death [26].

Similar to the aforementioned adipocytic differentiation, co-amplification of *CDK4* and *MDM2* occurs during mesenchymal stem cell differentiation towards osteoblasts as shown by FISH and qPCR analysis [15]. Amplification of *CDK4* and *MDM2* was found in 67% of parosteal osteosarcoma that is a low-grade differentiated type of osteosarcoma and in 9–12% of classical osteosarcoma [27]. Notably, mesenchymal stem cell and committed osteoblast precursors have been regarded as the cell of origin of osteosarcoma [28].

Amplification of *CDK4*, *MYO18B* and *NUP133* was detected by FISH and qPCR during the differentiation of mouse C2C12 myoblast cells, and amplification of *MDM2*, *CDK4* and *NUP133* during the differentiation of primary human myoblasts. Amplification of *CDK4* and *MDM2* was also found in leiomyosarcoma and rhabdomyosarcoma which are both myogenic sarcoma [29,30].

## 4. DNA Strand Breaks during Differentiation

Gene amplifications in tumors were often associated with genome instability including double strand breaks. Specifically, the induction of double strand breaks (DSBs) by gamma-irradiation can trigger gene amplifications [31,32]. In tumors endogenous DNA breaks initiate gene amplification during physiological stress including replication stress [33]. In a mouse model, ionizing-radiation induced DSBs were high-risk factors for development of glioblastoma with *MET* gene amplification [34]. As shown in the following there are also several studies reporting double strand breaks (DSBs) in stem cells and during stem cell differentiation.

The appearance of transient strand breaks has been reported during primary chick myoblast differentiation [35,36]. In detail, Farzaneh and colleagues investigated primary chick myoblasts in culture that first proliferate and then terminal differentiate visible as myoblast fusion and multinucleate syncytia [35]. After 53 h, when 50–60% of the myoblasts have fused the authors, detected the highest number of single strand breaks. The authors speculated that the DNA strand breaks in differentiating cells can be attributed to DNA repair deficiency but then detected a proficient repair in differentiating muscle cells of gamma-irradiation induced breaks. The authors hypothesized that the DNA breaks are indicative of genome mobility [35]. Likewise, developmental gene amplification during a restricted time window can also be regarded as indication of genome mobility. One of the limitations of these studies were that breakage sites were only detected in bulk DNA but not in single cells. Dawson and Lough used in situ nick-translation to localize DNA breaks in single nuclei during in vitro chicken myogenesis and found a decline of DNA breaks when differentiation proceeded and DNA breaks were repaired at terminal differentiation [36].

Notably, during adipogenesis of murine and human pre-adipocytes there is an up-regulation of double strand break repair activity due to an up-regulation of DNA-PKcs (DNA-dependent protein kinase catalytic subunit) expression [37]. During early adipocyte differentiation DNA-PKcs increases and thereby contributes to an increase in non-homologues-end-joining (NHEJ) DSB repair. This observation probably accounts for the decline of DNA breaks during progression of chicken myogenesis as described above.

Twenty years after the study of Dawson and Lough, Larsen and colleagues reported another example for a correlation between DSBs and differentiation during myoblast differentiation. They found an increase in double strand breaks after 12 and 24 h of differentiation and a decrease in double strand breaks after 48 to 72 h of differentiation [38]. In addition, they detected that caspase3/caspase-activated DNase induce double strand breaks that drive myoblast differentiation by targeting double strand breaks to specific genome regions. In contrast, treatment with neocarzinostatin, that induces indiscriminate double strand breaks and random DNA damage was not a trigger for differentiation [38]. Inhibition of caspase3 or reduction in caspase-activated DNase (CAD) expression resulted in a “dramatic” loss of double strand break generation and a block of myogenic differentiation program [38]. Another study on mouse and human myoblast differentiation detected double strand breaks during early differentiation and simultaneous TP53BP1 expression indicating double strand break repair [17]. In summary, DSBs are frequently observed during early differentiation and decrease with proceeding differentiation. They are obviously not the result of a deficiency in DSB repair but appear nevertheless essential for the differentiation program. To date, the appearance and reason for DSBs during differentiation remains unsolved. In the following, we refer to this problem and indicate a possible link between DSBs, differentiation and gene amplification.

## 5. Double Strand Breaks and Re-Replication

DNA replication is a tightly regulated process with two main steps, i.e., origin licensing in late mitosis and early G1-phase and origin firing during S-phase. The major mechanism to prevent relicensing during S-phase in mammalian cells is the inactivation of CDT1, which is one of the essential replication initiation proteins [39,40]. Early mitotic inhibitor 1 (Emi1) inactivation leads to degradation of inhibitors of CDT1 activity followed by massive re-replication [41]. Neelsen and colleagues investigated single-stranded DNA gaps (ssDNA) and DSB in association with re-replication. They suggest that a deregulation of replication origin firing, in their experiments by Emi1 depletion, leads to ssDNA gaps that persist in the template causing fork stalling and breakage of re-replication forks [40]. In their analysis DSBs occur after re-replication of DNA template with ssDNA gaps. In another study they report that even “mild” oncogene-induced replication stress may not be detected by cell cycle checkpoints [42]. The authors suggest that reactivation of replication origins in unrepaired ssDNA gaps can impair chromosome integrity [40]. The postulated fork collision model of Alexander JL and Orr-Weaver 2016 explains the generation of double strand breaks (DSBs) during re-replication [43]. In addition, it is very likely that single strand breaks (SSBs) are also generated during these events as summarized in Figure 2. Stark and Wahl 1984 postulated that the resulting onionskin-like structure is mitotically unstable and disintegrates to small circles or after several recombination events to intrachromosomal DNA [9]. During differentiation the breaks are probably necessary for removal of extra DNA, enabling of exonuclease cleavage, repair of open ends, and return to genome stability. Main methods for detecting re-replication were explained in Figure 1.

Interestingly in primary cells as BJ and IMR90 cells, CDT1 overexpression leads to limited re-replication at cellular replication origins with none of the replicating cells showing more than 4N DNA content [44]. In addition, SV40 T expressing BJ cells revealed multiple re-replication rounds, but only at SV40 origins. DSBs were observed in both cases but did not disturb cell growth. Truong and colleagues hypothesize that in primary cells with an intact ATR checkpoint, checkpoint activation limits re-replication levels to a minimal amount and repair mechanisms sufficiently remove amplified DNA and repair re-replication-associated DSBs [44].

In *Drosophila* ovarian follicle cells, the gene amplification of eggshell genes is enabled through the over-replication of two clusters of chorion genes [45]. Alexander and colleagues investigated this re-replication (over-replication) and reported that re-replication caused fork instability and DSBs at regions of fork collisions. They found that non-homologous end-joining (NHEJ) repairs forks during re-replication [46]. Notably, both studies, the one from Truong on primary cells with CDT1 overexpression and the one from Alexander on ovarian follicle cells link re-replication, DSBs and amplified DNA with DSB repair.

## 6. Gene Amplification and Re-Replication

As early as 1984, Stark GR and Wahl GM and Hamlin JL and colleagues postulated a model for gene amplification caused by unscheduled DNA replication [9,47]. This unscheduled replication led to an unstable set of replication bubbles that resembles the structure of an onionskin. Later this model was referred to as “onionskin”-model of gene amplification. The postulated structure has been confirmed by electron microscopic studies on Drosophila follicle cells that revealed multiple replication bubbles lying side-by-side [48] and is still basic for Drosophila follicle cell gene amplification analysis [49]. Stark and Wahl further postulated that breakage–fusion–bridge cycles might occur to heal chromosome breakage from unscheduled replication and unstable replication bubbles. Unscheduled replication is at the origin of both the “onionskin” model and the “breakage–fusion–bridge cycle” model. Both models can explain how these processes lead to the manifestation of gene amplification. Recently first evidence of re-replication during normal human myoblast differentiation was shown [50]. This study reported re-replication simultaneously with gene amplification and further postulate that re-replication enables gene amplification during differentiation of normal human cells.

## 7. Summary and Conclusions

In *Drosophila* gene amplification is associated with re-replication and double strand breaks. In human tumors gene amplifications, DSBs and re-replication have been reported but mostly in the context of cell cycle checkpoint failure. As mentioned above we recently reported re-replication in human myoblast cells during differentiation towards myotubes [50]. During the differentiation double strand breaks and gene amplifications can be detected at various time points and to a varying extent.

Healthy and “normal” stem cells have intact checkpoint and repair mechanisms to protect their genome integrity while using time-restricted re-replication and gene amplification for fast protein expression during differentiation (Figure 3). In contrast, stem and progenitor cells that harbor mutations affecting replication licensing, checkpoint and/or DNA repair mechanisms are thus not able to protect their genome integrity. It is legitimate to assume that stem cells with accumulated mutations transform to cancerous or pre-cancerous cells without the ability to limit re-replication to a level that can be repaired. Proceeding unlimited re-replication and gene amplification without sufficient DSB repair results in unrepaired DNA lesions and in turn in cell death or larger genome instability and tumorigenesis (Figure 4). We feel that it is of importance for stem cell research to know about the capabilities of stem cells during differentiation with the resulting consequences and potential risks.

## Figures and Tables

**Figure 1 cells-12-00148-f001:**
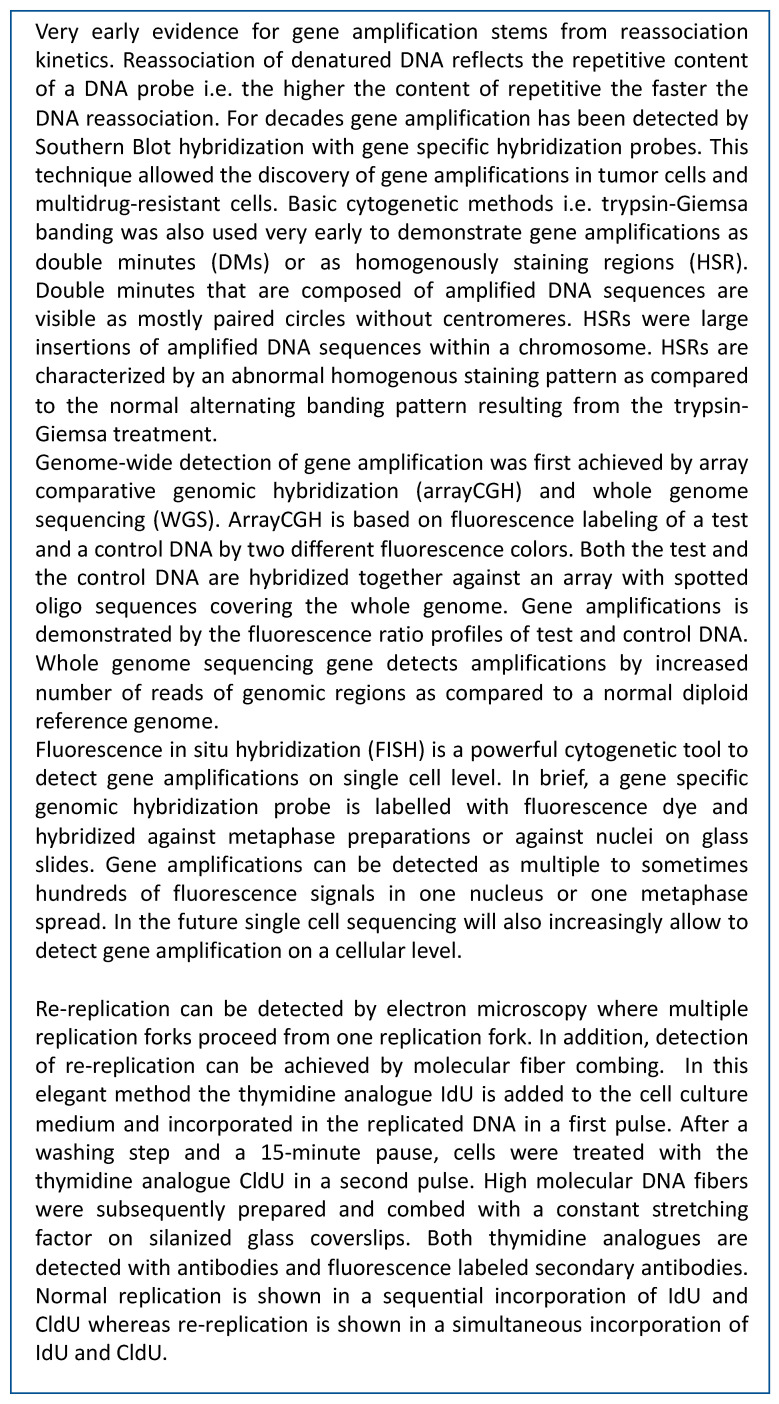
Main methods for detecting gene amplification and re-replication.

**Figure 2 cells-12-00148-f002:**
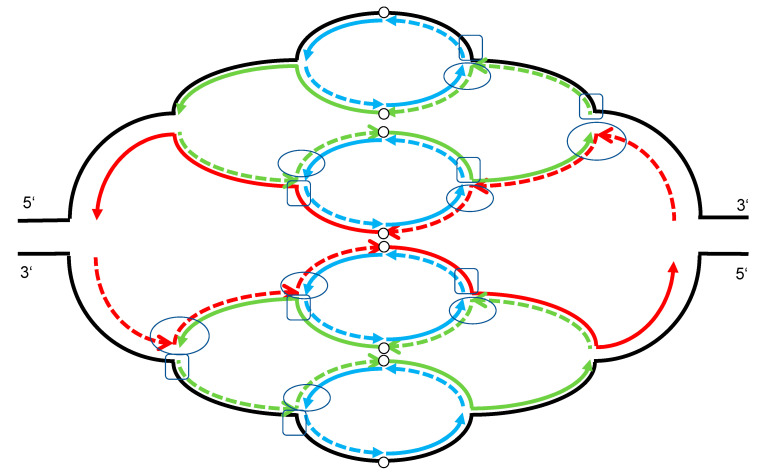
Double strand break (DSB) and single strand break (SSB) generation by re-replication. This scheme summarizes three rounds of replication with closed lines representing the leading strand and interrupted lines representing the lagging strand. Arrows indicate the synthesis direction of the replication. Origin of replications are indicated by open circles. The first round of replication is shown in red, the second in green and the third in blue. As proposed by Alexander [43] DSBs were generated when a leading strand from a second replication fork hits unligated Okazaki fragments on the lagging strand of an earlier fork as indicated by light circles. SSBs are very likely also generated in the lagging strand at this fork collision points as indicated by light rectangles.

**Figure 3 cells-12-00148-f003:**
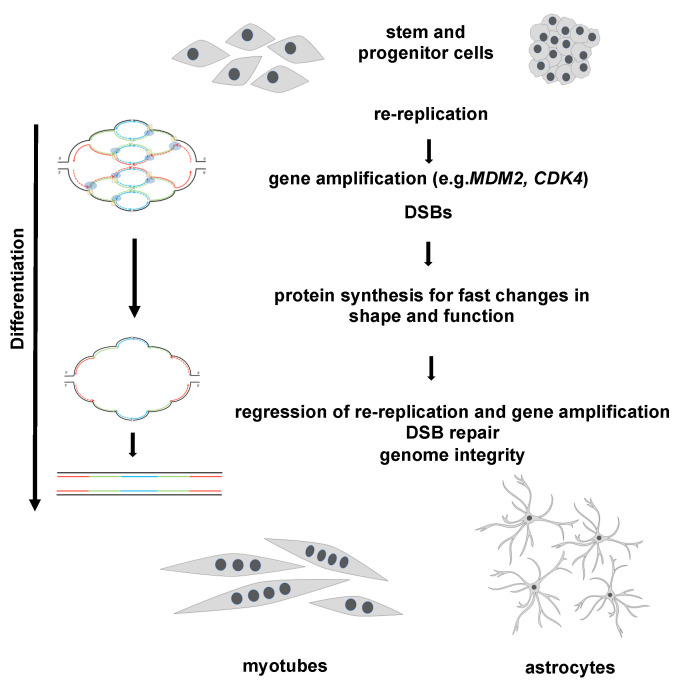
Transient re-replication and gene amplification during differentiation. The processes of re-replication and amplification during regular stem and progenitor cell differentiation are exemplarily shown for myoblasts and astrocytes. The re-replication causes gene amplification, which in turn enables increased protein synthesis. During the further differentiation both re-replication and gene amplification are ceased, and double strand break repair reinsures genome integrity. The scheme on the left-hand side explains how the onionskin-like structure can likely be resolved into duplicated DNA strands. Cell nuclei are indicated as filled dots.

**Figure 4 cells-12-00148-f004:**
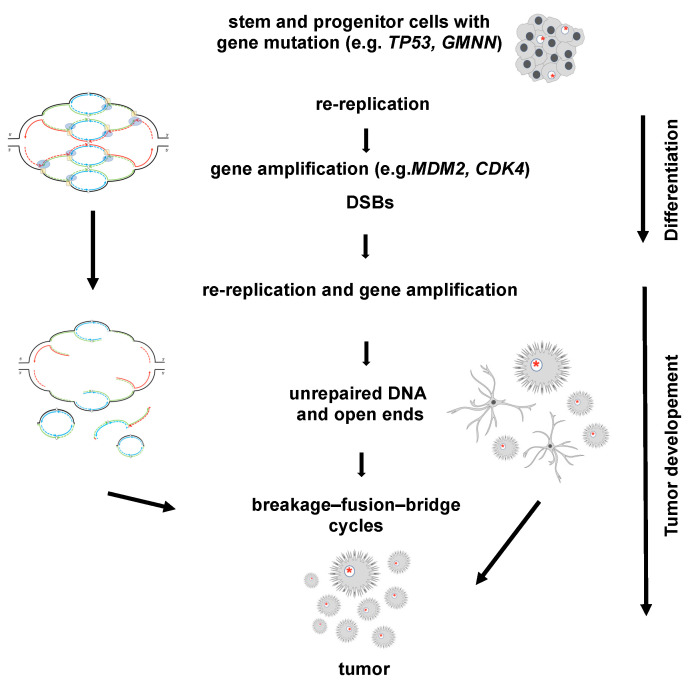
Hypothesized process on how stem and progenitor cells with gene mutations develop into tumor cells. The processes of re-replication and amplification leading to tumor development is exemplarily shown for astrocytes. As in the regular differentiation of stem and progenitor cells the re-replication causes gene amplification (see Figure 3). However, in the subsequent processes the gene mutations (indicated by red asterisks) of the stem and progenitor cells causing disintegration of the onionskin-like structure into small circles and open-ended DNA, which in turn lead to further gene amplification by breakage–fusion–bridge cycles as indicated by the schema on the left-hand side.

## Data Availability

Not applicable.

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
