# Peer review of "Gene Amplification in Tumor Cells: Developed De Novo or Adopted from Stem Cells"

_cells, 2022, doi:10.3390/cells12010148_

Round 1

Reviewer 1 Report

In this review article Fischer and Meese summarize findings about gene amplifications and re-replication as a normal part of differentiation, and speculate that the mechanisms are similar in tumorigenesis.

It is an interesting review that highlights the fact that many of the genes amplified in cancer are also found to be amplified during normal developmental processes. Yet the connection between this and re-replication is not clear.

I feel that it is necessary to be much clearer about the difference between rereplication and amplification. Are the authors trying to claim that rereplication is a normal process occurring in stem cells? If so, they provide only a single example for it (reference 46) and even when citing this paper, they do not say it explicitly. The authors should explain what are the differences between re-replication and gene amplification and clarify the evidence that exists for each phenomenon both in cancer and at early development.

Another issue that needs clarification is the model that they want to promote. Are they claiming that some of the amplified genes found in cancer were already amplified in early developmental stages and those cells that contain the amplification later evolved to be cancer (through selection). Alternatively, they may claim that there are certain biological processes that are used during normal development (i.e. re-replication of certain areas) and are adopted by the transformation process and get reactivated out of their normal context. While the first model seems to be hinted at by the title, the rest of the article is more consistent with the second model. This should be explained much more explicitly and the title should be changed accordingly.

The importance of re-replication in that context is not clear. In their 2022 paper (ref 46) they explained much more clearly that re-replication may be one of the mechanisms of gene amplification and the goal of their 2022 paper was to explore whether re-replication occurs during development. In the context of the review it should be better explained and ideally explored in the context of other mechanisms for gene amplification that may participate in cancer and in early development.

In addition, I think that the paper would benefit from a section that will describe the main methods that allow detecting gene amplification and re-replication and how one can distinguish between these two processes.

Finally, Figure 1 does not contribute to the paper since the illustration does not contribute to our understanding of what goes wrong during cancer transformation, only the captions and not the illustrations in the figure give a hint to what the authors want to convey.

Author Response

In this review article Fischer and Meese summarize findings about gene amplifications and re-replication as a normal part of differentiation and speculate that the mechanisms are similar in tumorigenesis.

It is an interesting review that highlights the fact that many of the genes amplified in cancer are also found to be amplified during normal developmental processes. Yet the connection between this and re-replication is not clear.

I feel that it is necessary to be much clearer about the difference between rereplication and amplification. Are the authors trying to claim that rereplication is a normal process occurring in stem cells? If so, they provide only a single example for it (reference 46) and even when citing this paper, they do not say it explicitly. The authors should explain what are the differences between re-replication and gene amplification and clarify the evidence that exists for each phenomenon both in cancer and at early development.

In the revised manuscript, we stronger differentiate between re-replication and amplification.  We clarify that re-replication is a process that is characterized by relicensing of replication origins during S-phase. By contrast amplification is the result of the re-replication process.  The difference between amplification and re-replication is also reflected in the different methods that are employed to detect either of the phenomena as detailed in the answer to the reviewer’s question below. We also clarify that the re-replication occurs both in tumors and in various non-tumor cells.

Another issue that needs clarification is the model that they want to promote. Are they claiming that some of the amplified genes found in cancer were already amplified in early developmental stages and those cells that contain the amplification later evolved to be cancer (through selection). Alternatively, they may claim that there are certain biological processes that are used during normal development (i.e. re-replication of certain areas) and are adopted by the transformation process and get reactivated out of their normal context. While the first model seems to be hinted at by the title, the rest of the article is more consistent with the second model. This should be explained much more explicitly, and the title should be changed accordingly.

The reviewer is right in that we would like to emphasis the second model.  We now clarify that re-replication is the underlying mechanisms that finally leads to the amplification of genes. The link between unscheduled DNA replication and gene amplification has been suggested as early as 1984 by Stark, Wahl, Hamlin and colleagues [9, 46].  As aforementioned, re-replication occurs both in tumors and in normal cells.  The difference between tumor and normal cells lies in the DNA structures that result from the re-replication. In normal cells an onionskin like structure was postulated and demonstrated for Drosophila follicle cells. By contrast breakage-fusion-bridge-cycle structures were postulated in tumor cells. Notably, both mechanisms result in gene amplification. This implies that tumor cells that have acquired amplified genes in this way, may also have acquired a growth advantage over the other tumor cells.

The importance of re-replication in that context is not clear. In their 2022 paper (ref 46) they explained much more clearly that re-replication may be one of the mechanisms of gene amplification and the goal of their 2022 paper was to explore whether re-replication occurs during development. In the context of the review it should be better explained and ideally explored in the context of other mechanisms for gene amplification that may participate in cancer and in early development.

As indicated above we now clarify in the manuscript that re-replication is considered as the main cause for gene amplification.  There are two hypotheses that explain how re-replication can lead to gene amplification.  First, unscheduled replication leads to an unstable set of replication bubbles that resembles the structure of an onionskin. That is why this model was later referred to as “onionskin”-model.  Second, unscheduled replication leads to chromosome breakages that are healed by breakage-fusion-bridge-cycles.  Re-replication is at the origin of both the “onionskin”-model and the “breakage-fusion-bridge-cycles” model.  Both models can explain how these processes lead to the manifestation of gene amplification.

In addition, I think that the paper would benefit from a section that will describe the main methods that allow detecting gene amplification and re-replication and how one can distinguish between these two processes.

We now included a new paragraph describing the methods that allow detecting either gene amplification or re-replication.  As detailed in the revised manuscript gene amplification can be detected by Southern Blot hybridization, by cytogenetic means including the detection of double minutes and homogenously staining regions, by fluorescence-in-situ-hybridization, by comparative genomic hybridization (arrayCGH), and by whole genome sequencing. Re-replication can be detected by electron microscopy or molecular fiber combing.  In the new paragraph we provide briefly describe these methods.

Finally, Figure 1 does not contribute to the paper since the illustration does not contribute to our understanding of what goes wrong during cancer transformation, only the captions and not the illustrations in the figure give a hint to what the authors want to convey.

We now included a new Figure 2 and revised the previous Figure 1 to new Figure 3 and Figure 4. Figure 2 visualizes re-replication and double strand/ single strand break generation. Figure 3 summarizes re-replication and gene amplification during normal differentiation of stem and progenitor cells whereas Figure 4 summarizes re-replication and gene amplification in gene mutated stem and progenitor cells leading to tumor development.  

Reviewer 2 Report

Fischer and Meese report an elegant review paper about gene amplification in tumor cells, questioning if these gene amplifications were acquired or de novo developed. The description of the different discoveries is fairly organized, although in certain cases there is the impression that some concepts are repeated in different paragraphs although not necessary. Overall the review is comprehensive of all the information to support the studied theme. A revision of the repeated concepts across the different paragraphs is advised.

Author Response

Reviewer 2

Fischer and Meese report an elegant review paper about gene amplification in tumor cells, questioning if these gene amplifications were acquired or de novo developed. The description of the different discoveries is fairly organized, although in certain cases there is the impression that some concepts are repeated in different paragraphs although not necessary. Overall the review is comprehensive of all the information to support the studied theme. A revision of the repeated concepts across the different paragraphs is advised.

We would like to thank the reviewer for the very encouraging comments. As suggested, we revised the manuscript and summarized the subsections dealing with the evidence for gene amplification into a single paragraph.  

Round 2

Reviewer 1 Report

The authors significantly improved the manuscript and it can be accepted in the present form.